# *MYD88* Wild Type in IgM Monoclonal Gammopathies: From Molecular Mechanisms to Clinical Challenges

Tina Bagratuni *, Alexandra Papadimou, Kostantina Taouxi, Meletios A. Dimopoulos and Efstathios Kastritis

Department of Clinical Therapeutics, School of Medicine, National and Kapodistrian University of Athens, 15772 Athens, Greece; apapadimou@med.uoa.gr (A.P.); taouxikon@med.uoa.gr (K.T.); mdimop@med.uoa.gr (M.A.D.); ekastritis@med.uoa.gr (E.K.)
* Correspondence: tbagratuni@med.uoa.gr

**Abstract:** High frequencies of $MYD88^{L265P}$ mutation are observed in IgM monoclonal gammopathies, and specifically in Waldenström macroglobulinemia (WM), indicating this mutation as a potential disease biomarker. Given the fact that $MYD88^{L265P}$ mutation has been described as a key driver mutation, has increased our understanding of the biology behind MYD88 signaling and helped us to identify the functional components which could be targeted. On the other hand, the absence of the $MYD88^{L265P}$ mutation in patients with IgM monoclonal gammopathies has been associated with a higher risk of transformation to aggressive lymphomas, resistance to several therapies, and shorter overall survival. The present review focuses on the molecular mechanisms that shape the signaling pattern in $MYD88^{WT}$ cells, as well as on the clinical implications and therapeutic challenges of WM patients that harbor the $MYD88^{WT}$ genotype.

**Keywords:** Waldenström macroglobulinemia; MYD88; wild type; IgM monoclonal gammopathies





## 1. Introduction

Mature B-cell neoplasms are clonal tumors of B-cells characterized as a group of diseases with a highly heterogeneous profile, both biologically and clinically. Depending on the entity, the clinical course may range from an indolent to an aggressive disease. Mature B-cell neoplasms constitute more than 90% of lymphoid neoplasms and, based on histology and immunophenotype, they account for 34 different entities, including diffuse large B-cell lymphoma (DLBCL), chronic lymphocytic lymphoma (CLL), Burkitt lymphoma (BL), lymphoplasmacytic lymphoma (LPL)/Waldenström macroglobulinemia (WM), splenic marginal zone lymphoma (SMZL), nodal marginal zone lymphoma (NMZL), mantle cell lymphoma (MCL), follicular lymphoma (FL), and hairy cell leukemia (HCL) [1]. They exhibit a broad spectrum of characteristic cytogenetic abnormalities and genetic aberrations, which are partly characteristic among different B-cell neoplasms but are (most of the time) not specific enough for a definitive diagnosis. Some of the cytogenetic abnormalities include recurrent translocations such as t(11;14) (q13;q32) seen in >95% cases of MCL, t(14;18) (q32;q21) seen in 90% cases of FL, t(8;14) (q24;q32) seen in 80% cases of BL, and 6q deletion (del6q) seen in 27% cases of WM [2–5] while genetic aberrations include gene mutations, such as *BRAF* V600E in HCL, immunoglobulin heavy chain gene (*IGHV*) in CLL, and *MYD88* L265P in WM [6,7].

IgM monoclonal gammopathy is a heterogeneous group of B-cell/plasma cell clonal diseases that includes a range of conditions from monoclonal gammopathy of undetermined significance to Waldenström macroglobulinemia, IgM multiple myeloma, and less common, other B-cell neoplasms secreting IgM.

Studies by Treon et al. and other researchers suggested the $MYD88^{L265P}$ mutation is present in >90% of WM, and that it could be important for the differential diagnosis of WM [8] vs. plasma cell malignancies. This mutation is also present in various other

B-cell neoplasms such as SMZL, CLL, and DLBCL, but at a lower frequency [9–13]. Studies have shown that WM patients lacking the $MYD88^{L265P}$ may be less responsive to Bruton's tyrosine kinase (BTK) inhibitors [14], which may also be associated with a lower number of tumor cells and lower International Prognostic Scoring System score at presentation [11]. In the most recent WHO nomenclature and classification, $MYD88$ wild-type ($MYD88^{WT}$) does not exclude the diagnosis of WM; however, it may be associated with a genetic profile other than $MYD88^{L265P}$ WM. Hence, the prognostic impact of $MYD88^{WT}$ genotype [11] requires further study. In this review, we aim to explore the latest findings on the $MYD88^{WT}$ genotype in various B-cell neoplasms, focusing on its role in tumor biology and its association with therapeutic challenges.

## 2. *MYD88*: Role, Pathway, Origin of Mutation

*MYD88* plays an important role in the functional integrity of the innate immune response. The *MYD88* gene was first described in the 1990s as a primary differentiation response gene which is upregulated during IL6-induced terminal differentiation and growth arrest. It encodes for a protein called myeloid differentiation primary response 88 (MYD88), located in the cytosol, which is involved in the signaling pathways within immune cells triggered by Toll-like receptors (TLRs) and interleukin-1 receptors (IL-1Rs). The MYD88 gene is located on human chromosome 3p22.2. It spans approximately 11.7 kilobases and consists of five exons [15]. In normal physiology, MYD88 acts as an adaptor of inflammatory signaling via the canonical NF-κB pathway. The MYD88 protein contains a death domain (DD), an intermediate linker domain (ID), and a Toll/IL-1 receptor (TIR) domain at the C-terminus. The DD enables protein–protein interactions; the absence of ID has been associated with the inability of MYD88 to support signaling [16] while the TIR domain mediates the downstream signaling cascade by interacting with TLRs and IL-1Rs. These domains are essential for MYD88's function in innate immune signaling [17,18]. Upon activation of TLRs or IL-1Rs, MYD88 is recruited to the receptor complex, leading to the formation of a signaling complex known as the Myddosome. This complex acts as a platform for the recruitment of downstream signaling molecules; activated MYD88 recruits IL-1 receptor-associated kinases (IRAKs), a serine-threonine kinase, and together they phosphorylate IRAK1 and IRAK2 which, in turn, interact with TNF receptor-associated factor 6 (TRAF6), initiating the activation of various signaling pathways, including transforming growth factor beta-activated kinase 1 (TAK1), mitogen-activated protein kinase (MAPK), and TAK1-binding protein (TAB) [18,19]. Activation of MYD88-dependent signaling pathways leads to the production of pro-inflammatory cytokines, such as tumor necrosis factor-alpha (TNF-α), interleukin-1 beta (IL-1β), and interleukin-6 (IL-6), as well as the expression of co-stimulatory molecules necessary for an effective immune response [19]. Ngo et al. were the first to identify that inhibition of MYD88 signaling via a non-synonymous, gain-of-function mutation in *MYD88* gene, leading to an amino acid change of leucine to proline at position 265 (NM_002468.5) (in the TIR domain), resulted in decreased NFκB activity and enhanced survival of activated B-cell-type diffuse large-cell lymphoma cell lines [12]. Other recurrent mutations in *MYD88* have also been reported, although the impact of these mutations is still under investigation due to their low prevalence [20]. As previously mentioned, MYD88 DD and ID are responsible for downstream signal propagation via IRAKs, whereas the TIR domain integrates signals from upstream TLR and IL1R [21–23]. In the case of the $MYD88^{L265P}$ mutation, the TIR domain of *MYD88*, where this mutation resides, is highly activated compared to the $MYD88^{WT}$, and this increases downstream signaling and formation of the Myddosome complex. It has been shown that mutated MYD88 recruits IRAK1 and, together with IRAK4, promotes the survival of activated B-cell-(ABC)-diffuse large B-cell lymphoma (DLBCL) cell lines [12]. Hence it has been hypothesized that $MYD88^{L265P}$ occurs in B-cell neoplasms where there is a strong selection for aberrant NFκB signaling [23]. Since $MYD88^{L265P}$ constitutively activates the NFκB pathway, it is contemplated as an important oncogenic driver in B-cell lymphomas [12,24,25]. Non-L265P mutations (M232T, S243N, S222R, and T294P) have an intermediate effect on

NFκB pathway signaling compared to the $MYD88^{WT}$, which shows the lowest activity [12]. In addition to NFκB activation, L265P induces B-cell proliferation, which is accompanied with the induction of TNFAIP3 [26]. However, although studies have shown that L265P triggers the anti-apoptotic NFκB signaling that, in turn, enables cell survival during B-cell development, is not capable of providing a continuous B-cell clonal selection on its own, and for this reason, a second somatic mutation is required [26]. In WM, these mutations usually reside in genes such as *CXCR4*, which is the second most mutated gene, *TNFAIP3*, *CD79 A/B*, and *ARID1 A/B* [27,28]. In WM, patients who harbor the L265P mutation have also been reported to bear a mutation in the 196 tyrosine residue of *CD79B* gene, leading to a better response to BTK-based therapies [29].

In addition to NFκB pathway signaling, the BCR pathway also plays an important role in B-cell survival and proliferation and the oncogenesis of various B-cell lymphomas in combination with *MYD88* mutations [30]. Within the BCR signaling cascade, BTK acts as an integral protein which forms complexes with $MYD88^{L265P}$ but not $MYD88^{WT}$ cells [31]. Furthermore, the level of phosphorylated BTK is higher in WM cells with L265P mutation than lymphoma cells with WT *MYD88* [31]. Therefore, inhibition of BTK would result in the disruption of the $MYD88^{L265P}$ complex but would not significantly affect the $MYD88^{WT}$ cells.

### 3. *MYD88* Mutation Detection Assays

Currently used methods to detect $MYD88^{L265P}$ mutation most often involve allele-specific polymerase chain reaction (AS-PCR), ddPCR and Sanger sequencing, or use of NGS-based protocols in unsorted or sorted (for Sanger sequencing or NGS) bone marrow (BM) aspirates of patients with IgM monoclonal gammopathies [32–38]. The sensitivity of the molecular assay for the detection of $MYD88^{L265P}$ should not exceed a detection limit of $10^{-3}$. It has been shown that conventional polymerase chain reaction (PCR) and Sanger sequencing–based methods for MYD88 mutational detection have a low sensitivity of 25%, and although fairly described, should be considered especially when used in non-selected B-cells [39,40]. Non-L265P *MYD88* mutations have also been identified in patients with WM, including S219C, M232T, and S243N [41,42]. Furthermore, the evaluation of cell-free DNA (cfDNA) for the mutational characterization and monitoring of disease burden has recently been described in several hematological malignancies, including IgM monoclonal gammopathies, and has shown remarkable results [43–45]. It is a less invasive, patient-friendly test that could provide a good diagnostic yield, even comparable to BM, but the challenges in the detection sensitivity should be evaluated. Data so far have shown that only highly sensitive techniques such as ddPCR or Cast-PCR should be used for the detection of $MYD88^{L265P}$ mutation in cfDNA [36,38]. However, all these techniques, although promising, need to be standardized and implemented in prospective studies before they can be used in clinical practice; therefore, the current recommendation for molecular analysis is to perform BM aspiration at diagnosis [3].

### 4. *MYD88* Mutation Status in B-Cell Neoplasms

L265P mutation was first reported in DLBCL [46]. The study by Ngo et al. found that *MYD88* mutations are more frequently seen in the activated B-cell-like subtype of DLBCL (ABC-DLBCL) at a frequency of 29% of cases, rather than the germinal center B-cell-like subtype (GCB-DLBCL) where the mutated cases are rare to absent [12]. The mutation is also frequent in the primary DLBCL of the central nerve system (70%), in primary cutaneous PCDLBCL leg-type (54%), and in testicular DLBCL (74%) [30].

Lymphoplasmacytic lymphoma (LPL) is a B-cell neoplasm characterized by the abnormal growth and clonal proliferation of small mature B lymphocytes and plasma cells in the bone marrow and lymphoid tissues. WM is a distinct clinical entity of LPL characterized by the presence of lymphoplasmacytic bone marrow infiltration and the secretion of monoclonal IgM immunoglobulin [1] WM represents about 95% of LPL cases, based on the presence of monoclonal IgM, while other types of LPL produce either IgA or IgG

monoclonal immunoglobulins (non-IgM LPL) [47]. Treon et al. ware the first to identify the presence of L265P mutation in 91% of WM patients compared to the frequency of 25% seen in non-IgM LPL [8].

The presence of disease-associated symptoms distinguishes the symptomatic from the asymptomatic/smoldering WM [48], whereas those patients with an IgM serum protein of less 3 g/dL and a BM infiltration of less than 10% but no symptoms, are classified as IgM monoclonal gammopathy of undetermined significance (MGUS) which is considered the pre-malignant phase of WM (or rarely IgM myeloma) [49,50]. Patients with IgM-MGUS are at higher risk for developing WM, DLBCL, and mucosa-associated lymphoid tissue (MALT) lymphoma, as well as chronic lymphocytic leukemia. $MYD88^{L265P}$ is found in 50–80% of patients with IgM-MGUS and cannot be used to differentiate WM (symptomatic or asymptomatic) from IgM-MGUS [32–35,51]. Studies have shown that IgM-MGUS patients with $MYD88^{L265P}$ are at greater risk of progression to WM [8,33,34,52,53] while the mutation has not been found in IgG or IgA MGUS [34,52]. On the other hand, in IgM myeloma, which is commonly characterized by the presence of t(11;14) translocation in clonal plasma cells, a typical cytogenetic feature of multiple myeloma (MM), $MYD88^{L265P}$ mutation is absent.

Marginal-zone lymphoma (MZL) is an indolent disease comprising 7% of all non-Hodgkin lymphomas [54]. There are three subtypes of MZL: MALT lymphoma, nodal marginal-zone lymphoma (NMZL), and splenic marginal-zone lymphoma (SMZL). Two small series report the presence of L265P mutation in 4–21% of MZL cases [1,9,55]. Interestingly, those with $MYD88^{L265P}$ were also more likely to present with an IgM paraprotein [9].

Finally, the $MYD88^{L265P}$ mutation seems to be absent in primary mediastinal B-cell lymphoma [12,20,56] and primary cutaneous follicle center lymphoma, and it is rarely present in hairy cell leukemia (1.1%) [10,57–59], Burkitt lymphoma (1.5%) [12], follicular lymphoma (1.9%), [23,57,60–62] and CLL (2.5%) [23,55,57,63–65].

## 5. $MYD88^{WT}$ Genotype in IgM Monoclonal Gammopathies

Patients with $MYD88^{WT}$ genotype have not been studied extensively due to the low prevalence of this genotype; hence, the effect of this genotype on the disease outcome of patients with IgM monoclonal gammopathies is still unclear. While most WM cases have mutated $MYD88$ gene, 5–10% do not carry $MYD88$ mutations. Some studies show that WT WM patients may have a shorter overall survival (OS) (10-year OS of 73% in WT versus 97% in mutated patients) [66,67] while other studies indicate that the OS is not affected in this subgroup of patients [28,68]. Treon et al. suggested that although $MYD88^{WT}$ patients with suspected WM fulfil the WHO criteria for WM diagnosis, around 30% have an alternate diagnosis [69] such as IgM MM, where the predominant plasma cell compartment and the high IgM levels are the main characteristics [70]. A study by Lee et al. showed that DLBCL patients with L265P had a statistically significant inferior overall survival compared to DLBCL patients with the WT genotype [57]. In other subtypes of B-NHL, such as CLL and SMZL, $MYD88^{L265P}$ is associated with superior survival compared to WT $MYD88$ [71–73]. In IgM-MGUS patients, although the presence of $MYD88^{L265P}$ mutation has been associated with greater risk of progression to WM [34,52], most IgM-MGUS patients never progress to WM or other lymphoproliferative disorders, so this mutation cannot be considered a unique pathogenic factor in WM, and other WM precursors might exist rather than the transformation from IgM-MGUS [51,74]. In contrast to the "classic" IgM-MGUS cases that typically evolve to WM or even MZL, IgM-MGUS cases with a plasma cell infiltrate, rather than a predominant B-cell clone, may serve as precursors to IgM MM [69]. A study by Treon et al. showed that among patients with suspected $MYD88^{WT}$ WM, 10% had findings consistent with IgM myeloma characterized by predominant plasma cell clone and a significantly higher IgM level compared to $MYD88^{WT}$ WM patients [69].

Few studies have compared the clinical and laboratory features of $MYD88^{WT}$ versus $MYD88^{L265P}$ cases in WM. Patkar et al. found that WT patients had lower hemoglobin and IgM paraprotein levels, lower tumor burden in the bone marrow, lower prognostic score,

higher total leukocyte counts (TLC), and higher platelet counts compared to *MYD88* mutated cases [11]. Treon et al., in a study which included 150 patients with B-cell neoplasms, also showed lower serum IgM levels, TLC, and bone marrow infiltration, but also an association with older age in WT patients. Given the low prevalence of WT genotype across patients with IgM monoclonal gammopathies, some experts in the WM field consider the disease with this genotype to be an entirely separate clinicopathological entity, distinct from the typical WM associated with *MYD88* gene mutation, and are proposing that the presence of the L265P mutation should be considered as a WM-defining feature [66]. However, since the WM disease characteristics and severity assessed by the IPSS-WM, the bone marrow involvement, and patients' performance status are similar between the two subgroups, the diagnosis could not be other than an active WM with a different genotype status. Therefore, more studies on $MYD88^{WT}$ patients need to be conducted, wherein the combination of high throughput molecular assays, such as single-cell RNA seq analysis and a close follow-up of these patients, will lead to a better understanding of this genetically distinct subgroup of patients at both the biological and the clinical level.

In terms of the genomic landscape of WM patients harboring the $MYD88^{WT}$ genotype, Hunter et al. provided the first—and, to date, only—study, aiming to explore the genomic and transcriptomic characteristics of WM WT patients in a cohort of patients that included 18 $MYD88^{WT}$ patients [75]. Data from this analysis were compared with previous genome and transcriptome data from $MYD88^{L265P}$ WM patients [8,42,69,76]. This analysis in WT WM patients identified the presence of somatic mutations in NFκB-related genes, in genes that impact epigenomic dysregulation, and in genes that impair DNA damage repair. Transcriptionally, $MYD88^{WT}$ patients showed similarities to the $MYD88^{L265P}$ patients, justifying the many overlapping disease characteristics noted between the two subsets of patients [66,69]. Transcriptomic studies have also shown that $MYD88^{WT}$ WM clonal cells represent an earlier stage of B-cell differentiation compared to the $MYD88^{L265P}$ clonal cells [76] which is also consistent with the lower rate of *IgH* somatic hypermutation previously described in $MYD88^{WT}$ WM patients [35].

WM patients with $MYD88^{WT}$ have also been shown to have an increased risk of disease transformation and resistance to ibrutinib monotherapy [69,75,77]. A study by Treon et al. showed a higher incidence of disease transformation to DLBCL in $MYD88^{WT}$ WM patients, which also contributed to 36% of the death events observed in these patients [69]. Furthermore, this study showed that associated DLBCL events in $MYD88^{WT}$ patients were also associated with shortened survival. In terms of response to ibrutinib therapy, IgM and hemoglobin responses were more frequent and deeper in $MYD88^{L265P}$ WM cases, and significantly lower in $MYD88^{WT}$ WM cases [78]. Response to therapy was also affected by the *CXCR4* mutational status, where patients with the $CXCR4^{WT}$ genotype achieved better response rates compared to those with the $CXCR4^{WHIM}$ genotype. Given the above data, it is suggested that patients with WT genotype should be followed closely due to the higher risk of histological transformation and higher resistance to BTK-based therapies [68,69,75,79,80].

## 6. $MYD88^{WT}$ and Related Genes

NGS data from Hunter et al.'s study on $MYD88^{WT}$ patients showed that the majority of the genes with distinct mutational patterns affected pathways of NFkB signaling, epigenomic regulators, and DNA damage response [75]. The mutations found in NFκB-related genes, which have also been found in aggressive B-cell lymphomas, were rare or absent in the L265P-mutated subset of patients, and included *TBL1XR1*, *PTPN13*, *MALT1*, *BCL10*, *NFKB1*, *NFKB2*, *NFKBIB*, *NFKBIZ*, and *UDRL1F* [81–83]. *TBL1XR1*, a gene frequently mutated in the WT patients, encodes transducin-β–like 1 X-linked receptor 1, which is a transcriptional regulator that interacts with nuclear hormone receptor corepressors [84] and which may play a regulatory role in the NFκB pathway and *Wnt*-mediated transcription. Deletions and mutations of *TBL1XR1* have also been reported in acute lymphoblastic leukemias [85,86], however, the specific mechanisms by which *TBL1XR1* mutations contribute to tumorigenesis are yet to be discovered. Furthermore, mutations in genes such

as *NFKBIB*, *NFKB2*, and *MALT1* were also identified in this study, i.e., genes that have also been linked to promoting ibrutinib resistance in MCL patients [87]. In terms of the mutations observed in epigenomic regulator genes closely linked to the $MYD88^{WT}$ patients, *KDM6A*, *KMT2C*, and *KMT2D* have also been found to be the most frequently mutated epigenetic regulators in several types of cancer [88–92] highlighting their role in tumorigenesis. On the other hand, structural alterations, such as the deletion of chromosome 6q (chr6q), mostly observed in $MYD88^{L265P}$-mutated WM patients (30–50%), seem to be absent in WT WM patients [41,42].

One other important mechanism in lymphomagenesis is the DNA damage repair (DDR) pathway, and studies have shown that lymphoma patients often display mutations in genes involved in DDR pathways [93–95]. DDR pathways have been shown to affect sensitivity to alkylating agents [96–99], and a study by Li et al. demonstrated that inhibition of PAK4 and NAMPT by KPT-9274, a compound which affects the DDR pathway, sensitizes WM cells to the activity of alkylating agents, such as melphalan or bendamustine [100]. So far, little is known about the role of *DDR* genes, and specifically mutations of the *TP53* gene, in the development of WM. About 8% of WM patients bear *TP53* mutations and studies have shown an association with poor survival [41,101,102] and an increase in frequency after the first-line of therapy [103]. *TP53* mutations are mostly associated with mutated *MYD88* and *CXCR4* [28,102,104]; however, Hunter et al. showed that $MYD88^{WT}$ patients are also presented with mutations in *DDR* genes, including *TP53*, and this subset of patients is considered to be an ultra-high-risk disease group [75].

Studies have shown that some WM patients with WT genotype can also harbor mutations that promote WHIM-like signaling in the *CXCR4* gene [42,105], although these almost always occur in those with a $MYD88^{L265P}$ mutation [42,66,105]. Although the frequency of $MYD88^{WT}/CXCR4^{MUT}$ WM patients is very low, it seems to also be accompanied with mutations affecting NFκB signaling [75,106]. Mutations in *NFkB*-related genes observed in a subset of $MYD88^{WT}$ patients are mainly observed downstream of the BTK pathway, involving genes such as *CARD11*, *BCL10*, *MALT1*, and *PTPN13* [75].

Finally, transcriptomic sequencing in WM patients revealed that gene expression in the $MYD88^{WT}$ patient cohort was quite heterogeneous compared to the $MYD88^{L265P}$ patient cohort, indicating a diversity in the pathogenesis of this population [76]. Specifically, a downregulation of genes associated with NFκB signaling was observed in these patients, including genes such as *IL6*, *IRAK2*, *TNFAIP3*, *NFKBIZ*, *TIRAP*, *PIM1*, and *PIM2*. In addition, Hunter et al. suggested that the upregulation of members of the PIK3 pathway observed in these patients, accompanied with increased promoter methylation, create a rationale for assigning PIK3 inhibitors and demethylating agents as targets for future preclinical studies [76].

## 7. $MYD88^{WT}$ and Therapeutic Implications

When it comes to therapy, there are several treatment options for patients with symptomatic WM, mainly based on monoclonal antibodies targeting CD20 (rituximab and newer ones) in combination with alkylators (cyclophosphamide, bendamustine) and, less often, 20S proteasome inhibitors (PIs) (bortezomib and carfilzomib) and BTK inhibitors (ibrutinib, acalabrutinib, zanubrutinib) [78,107–109]. IgM-MGUS patients, regardless of *MYD88* mutation status, usually do not require treatment [110]. Treatment of IgM MM usually includes regimens that are used for non-IgM MM patients [111]. Treatment of WM patients is highly personalized depending on their clinical features, preferences, and comorbidities, as well as the efficacy and toxicity profile of the various regimens [112,113].

The *MYD88* mutation status is also important given the interest in therapies targeting the components of pathways activated by the L265P mutation (Figure 1). As previously mentioned, MYD88 is preferentially complexed to phosphorylated BTK (pBTK) in WM cells harboring the L265P mutation, a complex which is observed less in lymphoma cells with the WT genotype. Furthermore, the same study shows that overexpression of MYD88 WT did not show enhanced BTK activation; hence, the use of ibrutinib, an inhibitor of BTK

kinase activity, resulted in decreased MYD88-BTK complexing in MYD88 L265P-expressing cells and not in MYD88 WT-expressing cells. Studies have shown that activation of BTK via signaling through the B-cell receptor or other signaling axes might contribute clinically relevant pro-survival signals in patients harboring the $MYD88^{WT}$ genotype [114]. The data from an extended follow-up study in relapsed/refractory WM patients, where ibrutinib was used as a monotherapy, showed that among patients with $MYD88^{L265P}$ mutation but no mutations in the *CXCR4* gene, responses rates were very high, with a 75% progression-free survival rate (PFS) at a follow-up of almost 5 years compared to and 3.5 years seen in patients with $MYD88^{L265P}$/CXCR4$^{MUT}$, and a median PFS of just 5 months in patients with $MYD88^{WT}$ genotype [115,116]. Additionally, no patient harboring the $MYD88^{WT}$ signature achieved partial response (PR) or better in this study [78]. The presence of rare non-L265P *MYD88* mutations does not seem to affect response to therapy using ibrutinib [14,113]. In terms of BTK inhibitors, the use of acalabrutinib, which is a more selective BTK inhibitor compared to ibrutinib, showed better response rates, but none of the patients achieved a very good partial response rate (VGPR) [117]. In another prospective study, the use of zanubrutinib, a potent second-generation BTK inhibitor, which has shown reduced off-target effects and a better BTK occupancy compared to ibrutinib, has shown better response rates in $MYD88^{WT}$ WM patients [118]. This study was part of the non-randomized arm of the ASPEN trial, comprising WM patients with only $MYD88^{WT}$ genotype, where zanubrutinib led to an outstanding 50% major response rate (MRR), including a 27% response rate with VGPR. Furthermore, its ongoing efficacy is highlighted by the fact that at 18 months follow-up, the median PFS and OS was not reached for these patients. The efficacy of BTK inhibitors in $MYD88^{WT}$ WM patients is indicated in Table 1.

**Table 1.** Data from BTK-based therapies on $MYD88^{WT}$ WM patients.

| | MYD88WT (*n*) | TN (*n*) | ORR | MRR | |
|---|---|---|---|---|---|
| Ibrutinib | 4 | 0 | 50% | 0 | Treon SP et al. J Clin Oncol 2020 [41] |
| Ibrutinib + Rituximab | 11 | | 82% | 73% | Dimopoulos MA et al. N Engl J Med 2018 [119] |
| Acalabrutinib | 14 | 1 | 79% | 64% | Owen R et al. Lancet Haematol 2020 [117] |
| Zanubrutinib | 26 | 5 | 81% | 50% | Dimopoulos MA et al. Blood Adv 2020 [118] |

Given the above data, knowledge regarding the *MYD88* and *CXCR4* mutation status of each patient seems to be important for the use of BTK-based therapy, especially in cases of ibrutinib monotherapy [78,120]; however, this may not be the case for all BTKis as data on non-covalent BKTis such as pirtobrutinib in patients with $MYD88^{WT}$ are still lacking [121]. It is notable, however, that data from the phase 3 iNNOVATE study indicate that the combination of ibrutinib and rituximab is not affected by the absence of *MYD88* mutations [122]. In both the iNNOVATE and ASPEN trials, the evaluation of *MYD88* mutational status was conducted centrally in NeoGenomics laboratory (NeoGenomics, Aliso Viejo, CA, USA).

Available data indicate that the presence (or absence) of $MYD88^{L265P}$ does not affect the efficacy of chemoimmunotherapy regimens (BR or dexamethasone, rituximab, and cyclophosphamide (DRC) +/− bortezomib) [123,124], thus, based on these observations, it seems reasonable to prioritize chemoimmunotherapy rather than BTKi monotherapy in WT WM patients. If BTKi therapy is considered, and if available, then second-generation BTKis (zanubrutinib, acalabrutinib) may be preferable over ibrutinib. Otherwise, a combination of ibrutinib with rituximab is an approved option, independently of *MYD88* mutational status, based on the subgroup analysis of the iNNOVATE study.

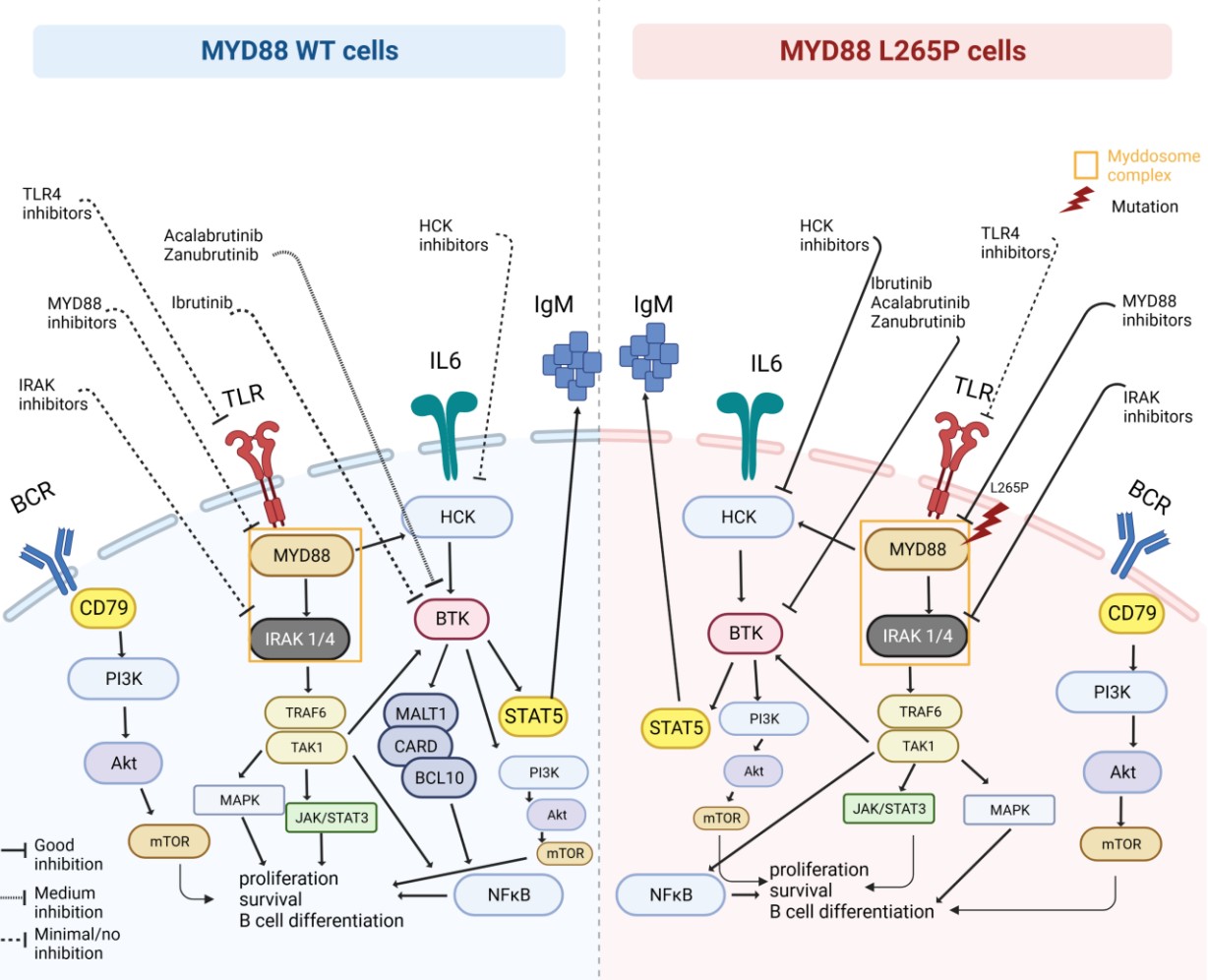

**Figure 1.** $MYD88^{WT}$ therapeutic challenges. Representation of the $MYD88^{L265P}$-mutated and $MYD88^{WT}$ cells and related pathways that can be targeted by specific drugs (created with BioRender.com, accessed on 7 August 2023).

Overall, BTKi-based therapies are more effective in $MYD88^{L265P}$ patients; however, second-generation BTKis seem to improve response rates in $MYD88^{WT}$ WM patients compared to the first-generation BTKi therapy. These ongoing studies need to be extended and more patients need to be included in order to obtain a clearer view of the efficacy of these therapies, especially in $MYD88^{WT}$ patients.

## 8. Conclusions and Future Perspectives

$MYD88^{WT}$ IgM monoclonal gammopathies have become a diagnostic and treatment challenge in the era of small-molecule targeting therapies. Data from studies have shown that patients with MYD88$^{WT}$ genotype appear to have a higher risk of transformation, shorter survival, and resistance to BTK-based therapy. Furthermore, the available data indicate that this is quite a heterogeneous group of lymphomas, which may include WM and its precursor conditions but also other lymphomas and plasma cell neoplasms. The genomic and transcriptomic data support the contention that WT WM and WM with $MYD88^{L265P}$ share some common characteristics despite their differences, and this is translated to a similar clinical course with most non-BTKi therapies. Thus, absence of MY88L265P should not exclude the diagnosis of WM. Ongoing research will further refine the special characteristics of $MYD88^{WT}$ WM/IgM monoclonal gammopathies by further dissecting the genetic characteristics of the clonal cells in an attempt to clarify the reasons behind the distinct clinical outcomes observed between the different BTK- and non-BTK-based

therapies in patients, and provide mechanistic insights as an opportunity to develop more personalized therapeutic strategies.

**Funding:** This research received no external funding.

**Institutional Review Board Statement:** Not applicable.

**Informed Consent Statement:** Not applicable.

**Data Availability Statement:** Not applicable.

**Conflicts of Interest:** The authors declare no conflict of interest.

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
