# Peer review of "MYD88 Wild Type in IgM Monoclonal Gammopathies: From Molecular Mechanisms to Clinical Challenges"

_hemato, doi:10.3390/hemato4030021_

Round 1
Reviewer 1 Report
The authors prepared an overview about an interesting subject, regarding MYD88 wild type IgM monoclonal gammopathies (= meant to include not only Waldenstroms Macroglobulinema).
Major comments:
1. IgM monoclonal gammopathies are defined as (line 38-40): "heterogeneous group of B-cell/plasma cell clonal diseases, that includes a range of conditions from Monoclonal Gammopathy of Undetermined Significance to Waldenström macroglobulinemia, IgM multiple myeloma and less common other B-cell neoplasms secreting IgM". However, chapter 5-6-7 only describes WM, not even WT IgM MGUS patients, and therefore the titel seems a bit misleading and not covering content. Can the authors add data on WT IgM MGUS? and what about IgM myeloma? A more clear line up of WT IgM monoclonal gammopathies would be helpful.
2. To be called a "MYD88 WT IgM monoclonal gammopathy", the choice of the test for the detection of the MYD88L256P mutation is extremely important. Could the publications referred to, miss some mutated patients, especially when Sanger was used on non-selected B-cells? And with the extremely low numbers, could this drive conclusions too prematurely? Chapter 3 describes some of the tests performed and this aspect should be added. In addition, there are some other MYD88 (non L256P) mutations described, such as S219C, M232T, and S243N and would be nice to add.
3. The authors raise an important subject regarding the definition of WM: should the MYDL256P mutation be a defining variable or not? But the reasoning thereafter (line 194-196) does not seem to be logically and needs to be better argumented.
4. The authors present mixed signals about the role of MYD88 WT WM and BTKi. There are some statements about the resistance when Ibrutinib is used, but later on it is clear that the data with the other BTKi and when combined with Rituximab this does not hold. This is important and should be more clearly explained and be consistent throughout the manuscript.
Minor comments:
- line 12; what is meant with "on the hand"?
- line 23; is a word missing from: "the clinical course may range from an indolent to aggressive"
- line 44; "but" : there is no contrast between disctintion of WM vs IgM myeloma and other B cell neoplasms so make 2 sentences
- line 46; BTK first use, explain abbreviation
- line 49; add that reference
- chapter 2; explanation of MYD88: is it present in all cells (line 57-58), where in the cell is the protein located (line 59), what is a signalling adaptor (line 63), what is pBTK (line 112)?
- line 244-45; "and patients with germline variations in one of the DDR genes should 244 be monitored for the development of lymphoma" is this true? Do these references state and provide evidence that indeed this is done? Are there (international) guidelines available?
- line 250; "TP53 mutations account for about 8% of WM patients..". please rephrase, meaning not clear.
English is not up to standard, see also minor comments
Reviewer 2 Report
The review by Bagratuni et al. on MYD88-wild type (WT) IgM monoclonal gammopathies is interesting, clear, well written, and the bibliography is complete and up-to-date.
Minor point: Since patients with MYD88-WT genotype appear to have a higher risk of transformation, shorter survival, and resistance to BTK-based therapy, perhaps this should be stated more clearly in the conclusion.
Round 2
Reviewer 1 Report
The authors improved their manuscript accordingly.
Cytocol is spelled Cytosol and critical evaluation of the text advised.